# Safety Evaluation and Population Pharmacokinetics of Camostat Mesylate and Its Major Metabolites Using a Phase I Study

**DOI:** 10.3390/pharmaceutics15092357

**Published:** 2023-09-21

**Authors:** Gwanyoung Kim, Hyun-ki Moon, Taeheon Kim, So-hye Yun, Hwi-yeol Yun, Jang Hee Hong, Dae-Duk Kim

**Affiliations:** 1Life Science Research Institute, Daewoong Pharmaceuticals, Yongin-si 17028, Republic of Korea; pharmrich@daewoong.co.kr (G.K.); 2220399@daewoong.co.kr (H.-k.M.); luluokj@gmail.com (T.K.); shyun389@daewoong.co.kr (S.-h.Y.); 2College of Pharmacy and Research Institute of Pharmaceutical Sciences, Seoul National University, Seoul 08826, Republic of Korea; 3College of Pharmacy, Chungnam National University, Daejeon 34134, Republic of Korea; 4Department of Bio-AI Convergence, Chungnam National University, Daejeon 34134, Republic of Korea; 5Department of Pharmacology, Chungnam National University College of Medicine, Daejeon 35015, Republic of Korea

**Keywords:** camostat mesylate, GBPA, GBA, population pharmacokinetics, modeling, simulation

## Abstract

Camostat mesylate is expected to be promising as a treatment option for COVID-19, in addition to other indications for which it is currently used. Furthermore, in vitro experiments have confirmed the potential of camostat and its metabolites to be effective against COVID-19. Therefore, clinical trials were conducted to evaluate the safety and pharmacokinetic characteristics of camostat after single-dose administration. Additionally, we aim to predict the pharmacokinetics of repeated dosing through modeling and simulation based on clinical trials. Clinical trials were conducted on healthy Korean adults, and an analysis was carried out of the metabolites of camostat, GBPA, and GBA. Pharmacokinetic modeling and simulation were performed using Monolix. There were no safety issues (AEs, physical examinations, clinical laboratory tests, vital sign measurements, and ECG) during the clinical trial. The pharmacokinetic characteristics at various doses were identified. It was confirmed that AUC _last_ and C_max_ increased in proportion to dose in both GBPA and GBA, and linearity was also confirmed in log-transformed power model regression. Additionally, the accumulation index was predicted (1.12 and 1.08 for GBPA and GBA). The pharmacokinetics of camostat for various dose administrations and indications can be predicted prior to clinical trials using the developed camostat model. Furthermore, it can be used for various indications by connecting it with pharmacodynamic information.

## 1. Introduction

Acute respiratory syndrome coronavirus 2 (SARS-CoV-2) has caused an ongoing pandemic [1]. The emergence of mutations in COVID-19 has further exacerbated the issues at hand. To date, there have been around 690 million reported cases worldwide, and the numbers are still increasing. Over the years, numerous pharmaceutical companies have made efforts to develop new vaccines and drugs for COVID-19 [2]. Additionally, drugs used for other indications were investigated for their potential application in the treatment of COVID-19 [3]. Nineteen drugs that have potential anti-COVID-19 activity were investigated by the US Department of Health and Human Services [4]. These drugs can inhibit the viral entry of host cells, viral replication, 3C-like protease, and viral RNA synthesis by targeting the activity of RNA polymerase. Clinical trials have been conducted using drugs such as favipiravir, pirfenidone, and ribavirin, either alone or in combination, to assess their effectiveness in treating COVID-19. Recently, the U.S. FDA approved Actemra (tocilizumab), Veklury (Remdesivir), Olumian (baricitinib), and Paxlovid (nirmatrelvir and ritonavir) as treatments for COVID-19, and 11 drugs were certified emergency use authorization (EUA)-authorized products.

Camostat mesylate is used in Korea for the treatment of chronic pancreatitis and reflux esophagitis after gastrectomy. It is utilized to inhibit TMPRSS2, a human cell-surface transmembrane serine protease. The protease is necessary for the activation of SARS-CoV lung infection as a key mediator of viral entry [5]. In vitro studies conducted on human lung cell lines and primary lung epithelial cells have produced promising results regarding the efficacy of camostat mesylate in blocking TMPRSS2-mediated viral entry [6]. Furthermore, administration to mice with a lethal dose of SARS-CoV-2 significantly reduced lethality by 60% [7]. Clinical trials on camostat mesylate have been conducted steadily since 2020, exploring its dosing, efficacy, and safety. Notably, clinical trials targeting early-stage or mild–moderate COVID-19 patients and comparing hospitalized and outpatient cases have shown improvements [8,9,10,11,12,13,14]. Furthermore, investigations have been carried out on the possibility of co-administration with other drugs [15,16]. 

After administration, camostat mesylate is rapidly converted to the active metabolite GBPA (4-(4-guanidinobenzoyloxy) phenylacetic acid, FOY-251) by carboxylesterase (Figure 1). Due to alkyl- and aryl-ester bonds, camostat mesylate exhibits high instability in blood [17]. Therefore, it is not metabolized by CYP enzymes or liver metabolism but is primarily influenced by esterase activity. Camostat and its metabolites are excreted into bile [18]. After the intravenous infusion of 14C-camostat mesylate in men, approximately 92.5% of the administered camostat mesylate was excreted in urine, with a minor amount of about 1.4% of the dose being eliminated through feces [19]. GBPA, the active metabolite, exerts inhibitory effects on COVID-19 through the same mechanism as camostat mesylate. In Calu-3 lung cell cultures, GBPA has shown an EC50 of approximately 178 nM [20]. Furthermore, biochemical assays revealed an IC50 value of about 4.3 nM for GBPA [21]. GBPA undergoes further hydrolyzation to the non-active metabolite 4-guanidinobenzoic acid (GBA) by arylesterase. Although GBA exhibits minimal inhibition of TMPRSS2, it can covalently bind to the enzyme [22,23]. Notably, the plasma concentration of GBA is higher than that of GBPA. Additionally, a portion of GBA remains unmetabolized and generates impurities [24]. To address potential safety concerns following the administration of camostat mesylate, we simultaneously analyzed both GBPA and GBA.

This study aims to investigate changes in the PK parameters and provide a safety analysis regarding a dose escalation of camostat from 100 to 300 mg in healthy Korean volunteers. Furthermore, our objective is to develop a model that can predict both GBPA and GBA simultaneously based on clinical trial data. This model will be utilized to predict the plasma concentration of GBPA and GBA during multiple dosing, allowing for the calculation of the accumulation index.

## 2. Materials and Methods

### 2.1. Study Design

A parallel-group, open-label study was conducted to compare and evaluate the safety and pharmacokinetics of different doses of camostat in healthy volunteers. Prior to the administration of the clinical trial drug, subjects who met the inclusion/exclusion criteria were assigned. Eligible subjects were healthy adults aged 19–55, weighing between 55.0 and 90.0 kg, with a BMI ranging from 18.0 to 29.9 kg/m^2^. They were required to meet the criteria for acute or chronic disease and internal medicine, and successfully pass a screening test, including clinical laboratory tests, a 12-lead electrocardiogram, and an assessment of vital signs. Non-pregnant women were also included as eligible participants. Subjects were excluded if they had any of the following conditions: clinically significant internal medicine disease, mental illness, acute illness symptoms within 28 days prior to the first administration of the drug, hypersensitivity to administered drugs or drugs of the same class, reflux esophagitis, lactose intolerance genetic disease, hypo- or hypertension, values exceeding the upper limit of AST and ALT, eGFR values less than 60 mL/min/1.72 m^2^, or tested positive for serological tests (RPR Ab, anti-HIV (AIDS), HBs Ag, HCV AB). If a numbered subject dropped out before the first administration, they were replaced by a preliminary subject. The participants received single-tablet doses of 100, 200, and 300 mg under fasting conditions. Camostat was taken with 150 mL of water in the morning.

### 2.2. Ethics

The protocol for this clinical study was approved by both the Ministry of Food and Drug Safety (MFDS) and the institutional review board (IRB number: CNUH2021-02-018-016) of the clinical trial center at Chungnam National University Hospital on 23 February 2021 and 18 February 2021, respectively. This study was conducted in accordance with the study protocols, Good Clinical Practice, and the Declaration of Helsinki. It was also registered on ClinicalTrials.gov (identifier: NCT04782505). All subjects provided written informed consent.

### 2.3. Blood Sampling

Venous blood samples of 5 mL were collected using EDTA-K2 tubes with 1 mL saline. The samples were obtained through an indwelling catheter inserted into the forearm at various time points: 0, 0.25, 0.5, 0.75, 1, 1.5, 2, 2.5, 3, 4, 5, and 6 h after each dose of camostat was administered. Subsequently, the collected samples were centrifuged at 3000 rpm for 10 min. The resulting isolated plasma was collected for further analysis.

### 2.4. Bioanalysis

GBPA and GBA were analyzed using the LC/MSMS system (Triple Quad 5500) (Sciex, MA, USA). GBPA was separated with the ACE 5 C18 column (4.6 × 150 mm, 5 µm), while GBA was separated with the CAPCELL PAK SCK UG80 S5 column (2.0 × 50 mm, 5 µm). Sildenafil served as the internal standard. The calibration curve range was 0.5 (the lower limit of quantification) to 1000 ng/mL for both. Weighted (1/x^2^) regression analysis was used to create the calibration curves.

For GBPA, 10 μL of sildenafil (1000 ng/mL) and 700 μL of methanol (0.05% formic acid) were added to 100 μL of samples, and the tube was vortexed at 2500 rpm for 20 min using a multi-tube vortex. After centrifuging well-sampled samples at 13,500 rpm for 5 min, 140 μL of the supernatant was transferred to a tube, and 60 μL of 10 mM ammonium formate (0.1% formic acid) was added and dissolved well in a multi-vortex for 1 min. A total of 150 μL of the supernatant was transferred to a vial and 10 μL was loaded into LC-MS/MS. For GBA, 10 μL of sildenafil (1000 ng/mL) and 300 μL of methanol (0.05% formic acid) were added to 100 μL of samples, and the tube was vortexed at 2500 rpm for 20 min using a multi-tube vortex. A total of 150 μL of the supernatant was transferred to a vial and 10 μL was loaded into LC-MS/MS.

The positive MRM mode ranged from *m*/*z* 314.2 to 145.1 for GBPA and 180.1 to 163.0 for GBA and 475.1 to 283 for IS, respectively. The isocratic protocol was used for the mobile phase ingredient A (10 mM ammonium formate (0.1% formic acid): B (methanol) = 45:55 (*v*/*v*) for GBPA, and A (40 mM ammonium formate (0.1% formic acid)): B (acetonitrile) = 50:50 (*v*/*v*) for GBA. The data were collected using SCIEX Analyst (Ver. 1.6.3).

### 2.5. Statistical Analysis of PK and Safety

The plasma concentration–time profiles of GBPA and GBA in each subject were analyzed using Phoenix WinNonlin Version 8.3 (Pharsight Corporation, Mountain View, CA, USA) with a non-compartmental method. Pharmacokinetic parameters such as C_max_, AUC_last_, AUC_inf_, T_1/2_, T_max_, and CL/F were evaluated. The linear-up log-down method was used to calculate the PK parameters. Dose proportionality for C_max_ and AUC_last_ was assessed by conducting ANOVA on dose-normalized C_max_ and AUC_last_ and performing a regression analysis of these values as a function of the dose using a log-transformed power model [25].

Safety was assessed in subjects who received at least one dose. Safety was assessed by adverse events (AEs), physical examinations, clinical laboratory tests (specific gravity, white blood cell, Albumin, Urine-HCG, anti-HIV (AIDS), cocaine, etc.), vital sign (SBP, DBP, pulse rate, and temperature) measurements, and ECG. Adverse events were monitored and characterized according to the criteria for assessing adverse reactions, including severity and causality, with the investigational medicinal product used in clinical trials. Clinical laboratory tests, vital signs, physical examinations, and 12-lead electrocardiogram results were analyzed and evaluated for their clinical significance. If changes in test values were determined to have clinical significance, they were recorded as adverse events and assessed following the criteria and methods for adverse event evaluation.

A statistical analysis was conducted related to pharmacokinetics and safety using SAS (version 9.4 SAS institute, Cary, NC, USA).

### 2.6. Population PK Model Development

A population pharmacokinetic model was developed using a non-linear mixed effect model in Monolix (version 2021R2; Lixoft SAS, 2021). R (version 4.2.2; R Foundation for Statistical Computing, R Core Team, Vienna, Austria) and RStudio (Version 2022.7.2.576; RStudio, Inc., Boston, MA, USA) were utilized to prepare a dataset and visualize model outputs. The stochastic approximation expectation-maximization (SAEM) algorithm was used to build a model [26]. To explore the validity of the model, several conditions (lag time, the transit model, the Michaelis–Menten method, linear clearance, etc.) were applied with different numbers of compartments (one, two, or three) [27]. Additionally, to enhance the robustness of the model, several variabilities were tested, including between-subject variability, additive error model, proportional error model, and combined error model. [28]. Covariate model building was carried out using the stepwise covariate method.

During model development, the model was selected as more appropriate when the objective function values (OFV) were lower. The model was evaluated using the visual predicted check (VPC), goodness-of-fit (GOF), diagnostic plots, and bootstrap [29].

### 2.7. Simulations

The pharmacokinetic simulations of GBPA and GBA were performed using the final developed population PK models. The plasma concentration–time values of GBPA and GBA were analyzed with Phoenix WinNonlin version 8.3 (Pharsight Corporation, Mountain View, CA, USA). The dosing regimen of the simulation was set to three times a day for 14 days, which was selected based on COVID-19 clinical studies [30]. The simulation involved 1000 subjects. To calculate the accumulation index of GBPA and GBA, we used Equation (1):(1)$$\mathrm{Accumulation}\;\mathrm{index}=\frac{1.0}{\left(1.0-exp\;(-Lambda\_z*Tau)\right)}$$
where Lambda_z is first-order rate constant and Tau is the dosing interval for steady-state data.

## 3. Results

### 3.1. Population

In order to determine the eligibility for clinical trials, various assessments, including screening tests, medical history, physical performance tests, physical examinations, and clinical laboratory tests, were conducted within 4 weeks prior to the scheduled clinical trial. The results of the screening tests are summarized in Table 1. A total of 15 male subjects were included and 180 samples were collected. All subjects were male, with a median age of 26 years, weight of 66.5 kg, and height of 174.2 cm. No subjects dropped out after camostat was administered. Camsotat doses of 100, 200, and 300 mg were administered to each group (*n* = 5).

### 3.2. Pharmacokinetic Analysis

The PK analysis was conducted on a total of 15 subjects, with 5 subjects per dose (N = 5 per dose). The bioanalysis was conducted on a total of 180 samples, and the suitability of the analytical results was assessed using quality control samples (low, intermediate, and high concentrations). In terms of precision and accuracy, over 67% of the samples met the criterion of being within 15% of the theoretical values. Moreover, for the same concentration level, more than 50% of the samples satisfied the requirement of being within 15% of the theoretical values. 

The plasma concentration–time profiles of GBPA and GBA are shown in Figure 2. The results of the non-compartmental analysis (NCA) for GBPA and GBA are summarized in Table 2.

It can be observed that the C_max_ of GBA is about 2.0 times higher than that of GBPA. Additionally, the AUC_0-6_ and AUC_inf_ of GBA are approximately 3.7 times higher than those of GBPA. No significant differences were found in the T_max_ (time to reach maximum concentration) and half-life values of GBPA and GBA among the different camostat dosages (100–300 mg). The T_max_ values for GBPA and GBA were 1 h (0.5–1.5) and 2 h (1.5–3), respectively. The apparent clearance (CL/F) values for GBPA and GBA were 704.4 L/h (405.5–926.6) and 152.1 L/h (88.42–218.8), respectively. In terms of distribution volume, GBPA is approximately twice as large as GBA.

All *p*-values for the dose-normalized PK parameters, including C_max_ (*p*-value, 0.612) and AUC_last_ (*p*-value, 0.997) for GBPA, as well as C_max_ (*p*-value, 0.225) and AUC_last_ (*p*-value, 0.105) for GBA (Figure 3), were found to be greater than 0.05. The dose linearity of C_max_ and AUC_last_ for both GBPA and GBA was assessed using a log-transformed power model. Regression analysis confirmed the linearity of the slope along with its corresponding 95% confidence interval (95% CI). The estimated slopes for GBPA’s C_max_ and AUC_last_ were 1.0066 (95% CI, 1.0041–1.0091) and 1.0057 (95% CI, 1.0034–1.008), respectively. For GBA, the estimated slopes were 1.0039 (95% CI, 1.002–1.0059) for C_max_ and 1.0038 (95% CI, 1.0021–1.0059) for AUC_last_. The confidence intervals for all PK parameters were within the range of 0.8–1.25. These findings confirm the dose linearity of both GBPA and GBA within the dosage range from 100 to 300 mg.

### 3.3. Safety Analysis

In the Screening and Post Study Visit, all safety evaluations were conducted with partial completion in D-1 (physical examination, clinical laboratory tests, adverse reaction confirmation) and D1 (vital signs, adverse reaction confirmation). Subjects who received at least one dose of camostat were included in the safety analysis. There were no adverse reactions collected in the safety analysis group after administration. Clinically significant changes were not reported in the laboratory tests (albumin, white blood cell, hematocrit, etc.), vital signs (blood pressure, pulse, and temperature of eardrum), physical examinations, and 12-lead electrocardiograms. The results mostly fell within the normal range, and even results that were outside of this range were considered clinically insignificant. Furthermore, serious AE/ADR did not occur.

### 3.4. Population PK Model Development

The concentrations of GBPA and GBA obtained during Phase 1 of this study were utilized for population pharmacokinetic (PPK) modeling. A population PK analysis was performed using all observations of GBPA and GBA from 15 subjects, comprising a total of 360 data points.

The population PK model was developed for GBPA and GBA simultaneously. To investigate the best model, OFV, relative standard error (RSE) and graphical model evaluation method (visual predicted check, GOF, and individual model-fitting) were used. The one-compartment disposition for GBPA and GBA with first-order absorption, lag time, the Michaelis–Menten kinetic model, linear elimination kinetics, and proportional error model was chosen to describe the PK profiles of GBPA and GBA (Figure 4).

To explain the inter-subject variability, parameters that can reflect the between-subject variability (BSV) were tested on all parameters. After that, this was applied to lag time, V_max_, K_m_, GBPA central volume of distribution (V1), clearance of GBPA, and clearance of GBA. There are no covariate effects in the PK parameters (height, weight, etc.), and the correlation between V1 and Vmax was reflected. The final model had the lowest OFV and maintained a low RSE for all parameters. Additionally, the model showed the highest level of visual accuracy in predicting the observed data. The parameters of the final model and bootstrap analysis are shown in Appendix A. The VPC and the goodness-of-fit for GBPA and GBA suitably predicted the final PK model (Figure 5 and Appendix A).

### 3.5. Simulations

Simulations were performed using the final developed model to assess the exposure of GBPA and GBA following multiple administrations. A dose of 200 mg of camostat was administered three times a day for 14 days. The PK profiles of GBPA and GBA for the first and last five doses are presented in Figure 6. The calculated accumulation index was 1.12 for GBPA and 1.08 for GBA.

## 4. Discussion

In this study, we investigated the safety and changes in the PK parameters in accordance with the dose of camostat in healthy Koreans. Additionally, the dose linearity of GBPA and GBA was calculated. Utilizing clinical trial data, we developed a population PK model that was capable of predicting GBPA and GBA simultaneously. Finally, we predicted the plasma concentration in multiple doses of camostat and calculated an accumulation index.

Several clinical studies on camostat mesylate for the treatment of COVID-19 were performed in various countries. This clinical trial was the first conducted on Koreans, and a high dose of camostat was administered to its subjects. The trial aimed to gather important information for the future development of an extended-release formulation and to assess linearity. For this purpose, we selected three different dosage strengths: a lower dosage of 100 mg, the target dosage of 200 mg, and a higher dosage of 300 mg.

In the clinical trial, men were enrolled at a 100% rate. Consequently, there were no female data in the dataset used for model development. Although it has been confirmed that there are no pharmacodynamic differences based on gender, there is a lack of evidence to assess pharmacokinetic disparities [31]. Therefore, the simulation results should be utilized for predictions related to males. To address these limitations, further research including females should be conducted to overcome this challenge.

The identified violations in the clinical trial were related to subject registration process errors, which were determined to have no negative impact on the subject’s safety and well-being, as the weight error range was within 10%. However, these subjects were excluded from the analysis group. The range of T_max_ for GBPA and GBA in each dose group showed no difference. With the median and average values for half-life, there were no differences in the range of T_max_ for GBA for each dose group. As a result, it was confirmed that the pharmacokinetic blood collection conducted in this clinical trial was appropriate.

The analysis of GBPA and GBA in the range from 0.5 to 1000 ng/mL in human plasma by LC-MS/MS confirmed sufficient sensitivity, linearity, and suitability in all batches. All of the tested samples in those batches were compliant with the acceptance criteria. In addition, the results of the specimen verification analysis for GBPA and GBA were also compliant with the reproducibility criteria. The LC/MS/MS method was validated in accordance with the bioanalytical method validation guidelines: Korea Ministry of Food and Drug Safety (2013.12) and Guidance for Industry: Bioanalytical Method Validation (FDA, 2018).

Previous studies have focused on the PK modeling of GBPA only and were conducted to prove the relationship between the camostat plasma concentration and the treatment of COVID-19 using the PKPD model [20,32]. This study presents a joint population PK model of GBPA and GBA for doses of camostat mesylate ranging from 100 to 300 mg. In the literature, the fraction of GBPA metabolized to GBA is known: on average, 10% of IV GBPA is excreted through the renal route [19]. Based on this paper, we could determine how much of the GBPA metabolized to GBA. We used the Michaelis–Menten method to describe the metabolism of GBAP to GBA. As a result, about 82% of the GBPA metabolized to GBA in our model (Appendix A).

Two kinds of linearity evaluations indicated that there was no significant difference in the PK parameters from the administered dose; however, we conducted linearity evaluations with a very small sample size and, due to this limited number, there is a significant risk of statistical generalization error. Therefore, prudential consideration should be given to the statistical results.

We conducted a simulation where 200 mg of camostat was administered three times a day for a duration of 14 days. The results demonstrated that both GBPA and GBA have a low accumulation index. Therefore, safety concerns related to the accumulation of GBPA and GBA are expected to be low.

The serine protease inhibitor is not metabolized by, nor does it act as an inhibitor of, the CYP system. In vitro results have also demonstrated that camostat mesylate does not inhibit CYP1A2, 2C9, 2C19, 2D6, and 3A4 enzymes [11,15,33]. As a result, it can be inferred that the potential for drug–drug interactions (DDIs) involving camostat mesylate and its metabolites is low when co-administered with other drugs.

Indeed, camostat mesylate not only holds potential for the treatment of COVID-19 but also shows promise in treating the protein loss enteropathy after Fontan surgery because camostat has a mechanism that blocks the digestive process by inhibiting proteolytic enzymes. Therefore, it is necessary to obtain an accurate understanding of camostat’s PK. Although they are used for various indications, both camostat and GBPA have low potential as perpetrators in drug–drug interactions [33]. To date, our model has been developed using pharmacokinetics data only, but we aim to create a rational pharmacodynamic model using various drug effect data, which will allow us to observe the interaction between pharmacokinetics and pharmacodynamics. We also plan to develop models for indications for which camostat mesylate may be used in the future.

## 5. Conclusions

In conclusion, camostat was well-tolerated in all cohorts. There were no significant differences observed in the half-life and T_max_ values of GBPA and GBA across all doses. The dose linearity for PK parameters was demonstrated for both GBPA and GBA. We successfully developed a population PK model capable of predicting GBPA and GBA concentrations concurrently. Furthermore, through our model, we were able to predict the extent of accumulation of GBPA and GBA.

## Figures and Tables

**Figure 1 pharmaceutics-15-02357-f001:**
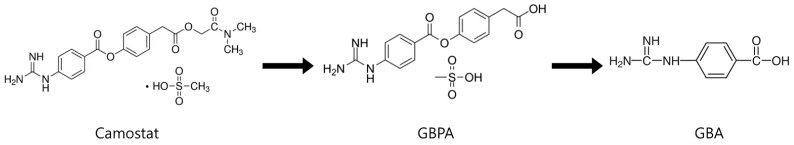
Metabolization of camostat mesylate.

**Figure 2 pharmaceutics-15-02357-f002:**
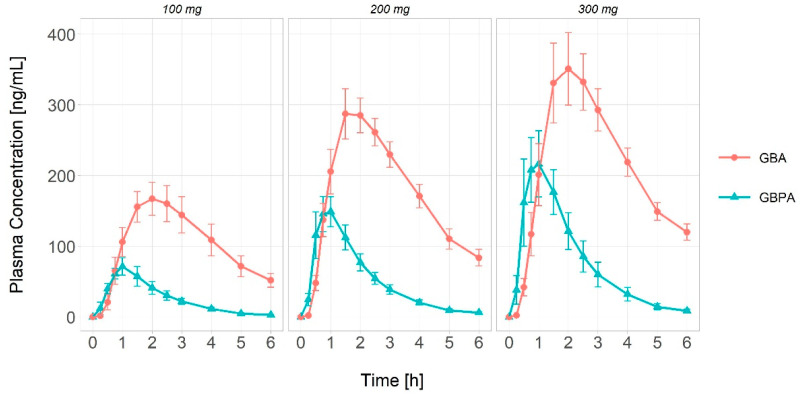
The plasma concentration–time profiles of GBPA and GBA after a single camostat administration. The error bars present the standard deviation. The blue lines are the GBPA plasma concentration and the red lines are the GBA plasma concentration.

**Figure 3 pharmaceutics-15-02357-f003:**
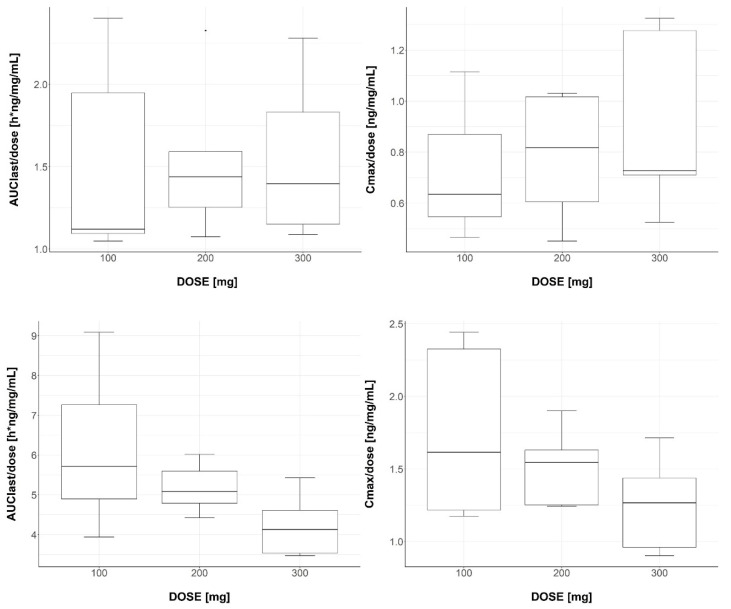
Boxplots of GBPA (**upper**) and GBA (**lower**). AUC_last_/DOSE and C_max_/DOSE after oral administration of 100, 200, and 300 mg of camostat. Median indicated by lines in boxes. The interquartile range box represents the middle, 50%, of the data. Whiskers represent the ranges of the bottom and top 25% of data values, excluding outliers.

**Figure 4 pharmaceutics-15-02357-f004:**
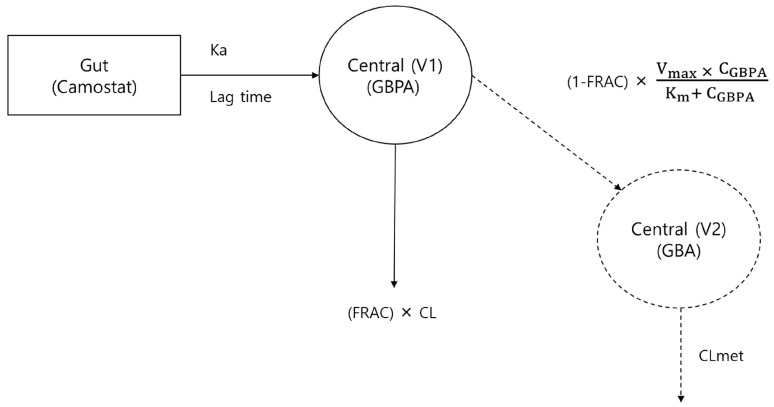
Schematic of population pharmacokinetic model for GBPA and GBPA. K_a_: first-order absorption rate constant, FRAC: fraction of GBPA to clearance, CL: clearance of GBPA, V_max_: Maximum velocity is the maximum rate of an enzyme catalyzed reaction K_m_: Michaelis constant, C_GBPA_: Concentration of GBPA, CL_met_: clearance of GBA.

**Figure 5 pharmaceutics-15-02357-f005:**
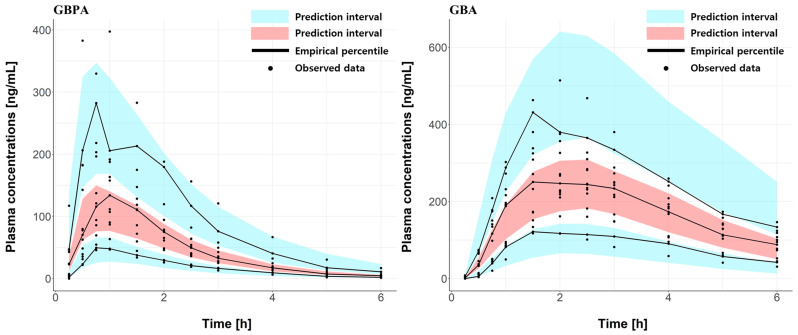
Final model of the visual perspective check for GBPA (**left**) and GBA (**right**). Black dots are observed data. Black lines are 95th, 50th, and 5th empirical percentiles of the predicted concentration. Blue areas represent the 95% prediction interval of the 95th and 5th. Red areas represent the 95% prediction interval of the 50th.

**Figure 6 pharmaceutics-15-02357-f006:**
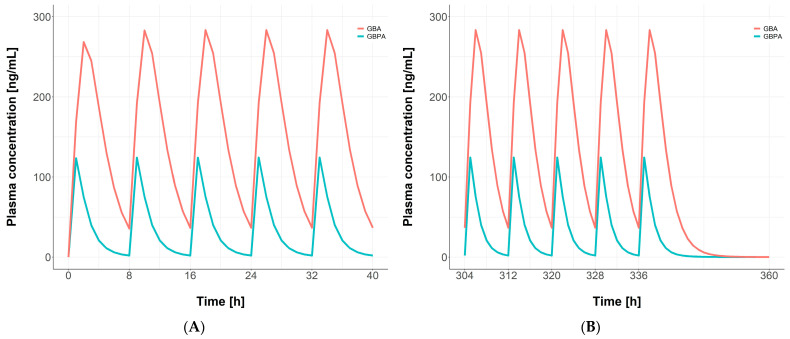
Simulation results of multiple doses of camostat. Averages for 1000 individuals for the first five doses of camostat (**A**) and last five doses of camostat (**B**) are shown. Blue lines are GBPA plasma concentration. Red lines are GBA plasma concentration.

**Table 1 pharmaceutics-15-02357-t001:** Demographic characteristics of subjects.

Variable		Statistics		
		100 mg	200 mg	300 mg
Sex, *n* (%)	Male	5 (100)	5 (100)	5 (100)
	Female	0 (0)	0 (0)	0 (0)
Age (years)		26.8 (6.34)	29.0 (5.56)	28.0 (5.34)
Weight (kg)		70.1 (10.9)	68.5 (10.9)	74.5 (16.0)
Height (cm)		173.1 (7.19)	172.8 (5.53)	175.4 (6.97)
BMI (kg/m^2^)		23.3 (2.60)	22.9 (2.93)	24.0 (3.96)
AST (IU/L)		17.4 (4.56)	16.6 (4.72)	13.6 (6.27)
ALT (IU/L)		24.2 (8.76)	24.8 (12.5)	21.8 (12.3)
eGFR (mL/min/1.73 m^2^)		99.2 (10.7)	113.8 (31.8)	93.8 (15.3)
Albumin (g/dL)		4.42 (0.21)	4.50 (0.12)	4.58 (0.29)

Values are given as the mean (standard deviation). BMI, body mass index; AST (IU/L); aspartate aminotransferase; ALT (IU/L), alanine aminotransferase; eGFR (mL/min/1.73 m^2^), estimated glomerular filtration rate; Albumin (g/dL).

**Table 2 pharmaceutics-15-02357-t002:** Pharmacokinetic parameters of GBPA and GBA (n = 5 per dose group) in healthy volunteers.

	GBPA			GBA		
	100 mg	200 mg	300 mg	100 mg	200 mg	300 mg
Half-life (h)	1.012 (0.122)	1.034 (0.083)	1.002 (0.058)	1.942 (0.377)	1.954 (0.565)	2.407 (1.089)
Cmax (ng/mL)	72.68 (26.42)	156.8 (50.85)	273.9 (108.8)	175.5 (60.07)	302.9 (55.26)	376.9 (101.1)
AUClast (h × ng/mL)	152.3 (63.68)	307.4 (96.45)	464.8 (150.5)	618.2 (203.0)	1036 (126.7)	1270 (243.6)
AUCinf (h × ng/mL)	156.5 (63.68)	316.9 (100.1)	477.5 (156.0)	762.8 (242.9)	1268 (194.5)	1710 (268.3)
CL/F (L/h)	141.7 (43.51)	158.8 (27.88)	179.6 (31.73)	718.6 (247.5)	675.1 (179.6)	680.4 (202.9)
Vd (h)	1046 (69.5)	1006 (286.8)	977.8 (265.4)	404.5 (163.5)	437.8 (86.23)	604.8 (207.2)

Values are given as the mean (standard deviation). AUC_inf_, mean total area under the plasma concentration–time curve from time 0 to infinity; AUC_last_, mean total area under the plasma concentration–time curve from time 0 to 6 h; C_max_, peak plasma concentration; half-life, terminal elimination half-life; V_d_, volume of distribution.

## Data Availability

Data is contained within the article.

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
