# Peer review of "Safety Evaluation and Population Pharmacokinetics of Camostat Mesylate and Its Major Metabolites Using a Phase I Study"

_pharmaceutics, 2023, doi:10.3390/pharmaceutics15092357_

Round 1

Reviewer 1 Report (Previous Reviewer 2)

In this resubmitted manuscript, the authors have resolved most of my previous concerns. The revised paper has been improved but there are still some issues that have not been addressed yet. Here are some remarks on the revised manuscript:

1. Follow my previous Comment 4: The detailed information on the safety evaluation and assessment should be added in the revised manuscript.

2. Again, if only male subjects were included in this clinical trial, I could not agree with your statement that "gender-related pharmacokinetic differences may not be considered clinically significant".

3. It is true that the authors did an analysis of variance with dose normalized PK parameters (AUC, Cmax) and power model to analyze dose linearity. However, it can't conceal the fact that this work only involved a quite small number of patients (N=15). I suggest the authors include this point as a limitation in the Discussion.

Moderate editing of English language required.

Author Response

Reviewer 2 Report (New Reviewer)

The topic of this manuscript is interesting and fits well the scope of the journal The reviewer feels it can be accepted after extensive amendments.

(1) The writing of this manuscript needs extensive polishing. For example, in the 2nd paragraph: The mechanism can help prevent ef-46 ficient degradation of the SARS-CoV-2 virus. It is misleading.

(2) The 1st sentence of the 1st paragraph is wrong. The information in the 1st paragraph is outdated, please amend with most recent information, particularly the drugs approved by authorities for the management of COVID-19.

(3) The chemical structures of the drug and its metabolites should be shown.

(4) The elimination pathway of GPBA / GBA should be mentioned.

(5) The details in bioanalysis such as accuracy, precision and stability should be provided.

(6) Does this drug has DDI with other drug(s)? The authors should discuss.

The topic of this manuscript is interesting and fits well the scope of the journal The reviewer feels it can be accepted after extensive amendments.

(1) The writing of this manuscript needs extensive polishing. For example, in the 2nd paragraph: The mechanism can help prevent ef-46 ficient degradation of the SARS-CoV-2 virus. It is misleading.

(2) The 1st sentence of the 1st paragraph is wrong. The information in the 1st paragraph is outdated, please amend with most recent information, particularly the drugs approved by authorities for the management of COVID-19.

(3) The chemical structures of the drug and its metabolites should be shown.

(4) The elimination pathway of GPBA / GBA should be mentioned.

(5) The details in bioanalysis such as accuracy, precision and stability should be provided.

(6) Does this drug has DDI with other drug(s)? The authors should discuss.

Author Response

Reviewer 3 Report (New Reviewer)

Camostat mesylate has been approved for treatment of pancreatitis in Japan and Korea, and is currently being repurposed for COVID-19 treatment. After administration, Camostat mesylate is rapidly converted to the active metabolite GBPA (4-(4-guanidinobenzoyloxy) phenylacetic acid, FOY-251) by carboxylesterase. GBPA undergoses further hydrolyzed to the non-active metabolite GBA (4-guanidinobenzoic acid) by arylesterase.

Since the pharmacokinetics of GBPA and GBA are not known, the authors simultaneously analyzed GBPA and GBA on voluntary patients, which represents new information about this drug.

Minor.

The authors mention the functional groups of the investigated drug and metabolites, therefore it is desirable to show the structure of Camostat mesylate and the scheme of its metabolism, i.e., formation of GBPA and GBA.

For equation (1) - accumulation index of GBPA - the parameters of the equation are not explained, a more detailed explanation of equation (1) and how the parameters of equation (1) were obtained is needed.

Round 2

Reviewer 1 Report (Previous Reviewer 2)

The authors have addressed my concerns in this revision. I have no more comments.

Moderate editing of English language required

This manuscript is a resubmission of an earlier submission. The following is a list of the peer review reports and author responses from that submission.

Round 1

Reviewer 1 Report

1. What were the inclusion/exclusion criteria to the study?

2. Was the LC/MS/MS method validated? According to what criteria?

3. In my opinion, the demographic characteristic of the subjects with the division into the groups taking 100mg, 200mg and 300mg of the drug should also be included.

4. In table 2 the following data should be included: volume of distribution and the absorption rate constant.

5. The analysis of the graphs in the figure 1 indicates that the sampling was ended too fast. It is seen especially for the dose 300 mg.

6. The quality of the figure 4 is poor. It is hard to read the description in the figure.

7.In line 187 the following sentence should be corrected: "The Cmax of GBA is about 2 times higher than the Cmax of GBA".

Reviewer 2 Report

This manuscript by Kim et al investigated changes in the pharmacokinetics parameters and provided a safety analysis regarding a dose escalation of 100 to 300 mg for the oral administration of camostat mesylate to healthy Korean volunteers. The authors built a population PK model that can predict GBPA and GBA concentrations simultaneously and confirmed that the accumulation hardly occurred during repeated administration. Overall, the conclusion can be supported by the current results, but there are still some critical issues that need to be addressed before further consideration.

1. The introduction should be intensively improved since the literature review fails to mention the new and growing literature on camostat mesylate in healthy adults for the treatment of COVID‐19.

2. After going through the whole manuscript, I wonder why do you prefer to study the dose of 100 to 300 mg in this Phase I study?

3. Line 74: “The subjects who met the selection/exclusion criteria were given a subject number before the administration of the clinical trial drug.” Please add the detailed inclusion criteria and exclusion criteria of the clinical trial.

4. 2.5 PK and safety analysis: Please specify when the safety was assessed in subjects who received at least one dose. One day or one week or some other time after the last dose?

5. 3.1 Population: Please justify why only male subjects were included in this clinical trial? If so, the conclusions of the study may not be convincing.

6. What is the half-maximal effective concentration of GBPA in this study? 

7. Actually, this work involved a quite small number of patients (N=15), and I even can’t find any statistical analysis or statistical significance. Therefore, these findings reported should be considered with extreme caution.

Moderate editing of English language.